# Predictors of Change in Stepping Stones Triple Interventions: The Relationship between Parental Adjustment, Parenting Behaviors and Child Outcomes

**DOI:** 10.3390/ijerph192013200

**Published:** 2022-10-13

**Authors:** Matthew Sanders, Nam-Phuong T. Hoang, Julie Hodges, Kate Sofronoff, Stewart Einfeld, Bruce Tonge, Kylie Gray

**Affiliations:** 1Parenting and Family Support Centre, The University of Queensland, Brisbane, QLD 4072, Australia; 2Brain and Mind Centre, University of Sydney, Camperdown, NSW 2006, Australia; 3Centre for Developmental Psychiatry and Psychology, Monash University, Clayton, VIC 3800, Australia; 4Centre for Educational Development, Appraisal, and Research, University of Warwick, Coventry CV4 7AL, UK

**Keywords:** mechanism of change, developmental disability, evidence-based parenting, Triple P

## Abstract

The current study explored the process of change in Stepping Stones Triple P (SSTP) using a community-based sample of 891 families of children with developmental disabilities (DD) who participated in an SSTP intervention at a community level. A preliminary analysis of outcome data indicated that SSTP intervention was effective in reducing parental adjustment difficulties, coercive parenting, and children’s behavioral and emotional difficulties immediately after the intervention. The effects were maintained at 12-month follow-up. The results also indicated that change in parental adjustment over the course of intervention was significantly associated with a change in parenting behaviors. However, change in parenting behaviors but not change in parental adjustment, predicted children’s behavioral and emotional problems following the intervention. The results suggest that positive parenting skills are the most salient ingredient driving the change in child behaviors in SSTP interventions.

## 1. Introduction

### 1.1. Relations between Parental Adjustment, Parenting Behaviors and Children’s Outcomes in Families of Children with DD

Research into children with developmental disabilities (DD) has consistently pointed to the elevated risk of the development of behavioral and emotional problems among children of this group [1]. As reported in a recent meta-analysis, the rate of behavioural problems in children with DD is two to three times higher than in typically developing children [2,3,4]. Different factors can contribute to the development and exacerbation of behavioral and emotional issues within this population. Parental stress has consistently been identified as one of the most prominent contributors. Neece et al. [5] followed two groups of children (144 were typically developing children and 93 were diagnosed with a type of DD) from age three to nine years. Their cross-lagged panel analyses indicated that behavioral problems and parental stress covaried across time, but parental stress consistently arose as a predictor of child behavior problems while the effect of early child behavior problem on parental stress was much less consistent. This finding was supported by Lin et al. [6] study which examined the transactional relations between parenting stress and both internalizing and externalizing behavioral problems in 75 young children with ASD over 1.5 years. The findings also indicated that early parenting stress was significantly associated with later children externalizing problems.

While parental stress directly affects children’s emotions and behavior [7], it also influences parenting which may in turn further exacerbate children’s behavioral and emotional difficulties. When parents are distressed, they are less likely to be sensitive to their children’s behaviors, they might also be more likely to engage in irritable transactions and poor disciplinary practices (inconsistency, coercion) that reinforce undesirable behaviors in children, thus making them more likely to occur again [8,9]. Totsika et al. [10] analyzed data from 555 families of children with DD and found that early parental distress (at nine months) significantly predicted child behavioral and emotional problems at both age seven and age 11 years. This relationship was mediated by adversarial parenting practices between ages 3 and 5 years. Day et al. [11] also surveyed 1392 families of children with a disability aged between 2 and 12 years and found that parental adjustment difficulties (depression, anxiety and stress) were among the strongest predictors of coercive parenting.

Due to the reciprocal transaction between parental stress, parenting behaviors and children’s behaviors, it is not uncommon for parents of children with DD who exhibit elevated behavioral problems to also experience a high level of stress and use more coercive parenting. Interventions to reduce behavioral problems in children with DD thus commonly aim to address both parental stress and dysfunctional parenting behaviors [12].

### 1.2. Stepping Stones Triple P

A number of different parenting programs have been shown to be effective in managing behavioral problems in children with DD. Among those, Stepping Stones Triple P (SSTP) [13]—(a variant of the Triple P Positive Parenting Program) is one of the most extensively studied and widely used. Built on social learning theory, SSTP recognizes the reciprocal nature of parent–child interactions surrounding dysfunctional behaviors [13]. Therefore, the occurrence of child behavioral and emotional problems in children with DD is viewed as both a consequence and an antecedent of dysfunctional parenting and parental adjustment difficulties. In SSTP, the goals are to help parents learn to manage their children’s behavior problems without using coercion escalation or harsh discipline and to adopt better strategies to regulate their emotion [13].

Throughout the SSPT program, parents are encouraged to choose their own goals, develop plans and execute their plans which includes the capacity to plan and anticipate, regulate their emotions, solve problems, and collaborate with significant others to provide care for their children. This self-regulatory approach is expected to help reduce parents’ use of coercive and punitive disciplining and promote parents’ capacity to regulate their emotions and behavior throughout intervention [13]. In addition to the core Triple P strategies, the program incorporates a number of additional disability-related components to reflect the additional challenges faced by parents of children with disabilities as well as a focus on community living and family support movements (such as Being part of the community). These include: (1) Identifying additional factors that are more likely to contribute to the development of behavior problems in persons with disabilities (e.g., the accidental reward for stopping disliked activities). (2) Incorporating other behavior change strategies from the disability literature (such as setting up an activity schedule). (3) Developing additional protocols to deal with self-injurious behavior, repetitive behaviors, and pica that are more prevalent among children with disabilities. (4) Modifying wording and examples in parenting materials to make them more acceptable and sensitive to parents of children with disabilities [13].

Meta-analyses of randomized controlled trials of SSTP have consistently shown that SSTP effectively reduces harsh parenting, child behavioral and emotional problems, and parental distress. For example, Tellegen and Sanders [14] analyzed both controlled and uncontrolled design studies. They found that SSTP had moderate to large effects in reducing coercive parenting behaviors, moderate effects in reducing child behavioral and emotional problems, and moderate effects in reducing parental adjustment problems. In a recent systematic review and meta-analysis of all SSTP levels, Ruane and Carr [15] found that SSTP has small to medium effects on parental adjustment and co-parenting. For parenting behaviors and child behaviors, the effect sizes were medium to large. The growing literature supports the efficacy of SSTP in reducing harsh parenting, child behavioral and emotional problems, and parental adjustment difficulties. Yet, little is known about the process of change explaining the effects of SSTP for families of children with DD. It is not yet understood if a change in parental adjustment over the course of the intervention assists change in parenting behaviors and vice versa. It is also yet to know if changes in parental adjustment or changes in parenting behaviors contribute to the change in children’s behaviors and emotional outcomes as proposed by the literature.

### 1.3. Current Study

The present study examined the process of change in SSTP interventions, explicitly emphasizing the bidirectional association between change in parental adjustment and change in coercive parenting over the course of the SSTP intervention and their association with children’s subsequent outcomes. Specifically, we examined (1) how parents’ experience of emotional difficulties (parental adjustment) and use of coercive parenting affect one another before, during, and after the intervention and (2) how the change in parental adjustment and use of coercive parenting during the intervention predict subsequently reported a decrease in child behavioral and emotional problems. We hypothesized that: (1) the decrease in parental adjustment difficulties over the course of intervention will be associated with the simultaneous decrease in coercive parenting behaviors and (2) the decrease in parental adjustment difficulties and coercive parenting from pre- to post-intervention will predict a subsequent decrease in child behavioral and emotional problems at follow-up.

## 2. Materials & Methods

### Participants

Participants in this study were 891 parents and caregivers living in the Australian states of Victoria and Queensland and enrolled in the Mental Health of Young People with Developmental Disabilities (MHYPeDD) research study. These were caregivers of a child aged between 2 and 12 years who were recruited via a variety of pathways, e.g., posters, brochures, and newsletters prepared by the project team and disseminated by their current service provider, a project-specific Facebook page, direct contact from the project team or via their child’s school. Interested parents then provided evidence of a of diagnosis of DD provided by a suitable professional such as a psychiatrist, psychologist, speech pathologist, neurologist or pediatrician. Although most parents responded to questionnaires online, there was also the option of telephone interviews or hard copies if needed. Most participants (87.65%) were mothers (biological, stepmother, adoptive mother) of at least one child with DD. The majority of target children were male (77.10%) aged between 2 and 12 years old (*M =* 4.98, *SD =* 2.65). At the time of enrolment, 76.77% parents reported their child also had a diagnosis of ASD (Table 1).

## 3. Procedure

The study received ethical clearance from the Behavioral and Social Sciences Ethical Review Committee at the University of Queensland. Professionals involved in the MHYPeDD project received training on at least one SSTP program, and they participated in a two-year implementation period to deliver interventions to families of children with DD. Parents were referred by their current service providers, directly via the SSTP project team (including the Facebook page) or their child’s school to attend either Primary Care SSTP (3–4 brief individual sessions), SSTP seminars (120-min large-group presentations), Group SSTP (5 group sessions and follow-up telephone calls), Standard SSTP (10 individually delivered sessions); Self-directed Triple P or Triple P Online (self-administered). Parents completed a short package of measures before the intervention, after the intervention and then at six months following their attendance.

To minimize site differences, all practitioners received identical competency- and accreditation-based training and all interventions were delivered with the same practitioners and parent resources. This is a widely used method of Triple P dissemination, ensuring fidelity to the program. Studies of Triple P regular service delivery have shown that there are few differences between training outcomes for practitioners from different disciplines, countries, and levels of programs [16,17].

## 4. Measures

### 4.1. Demographics

Demographic variables used for analysis in the present study included the child’s age and gender, type of DD (with or without ASD), caregivers’ relationship to the child, financial hardship and level of intervention. Responses were mainly based on the primary carer. Financial hardship was assessed using the question: “*Suppose you only had one week to raise $2000 for an emergency. Which of the following best describes how hard it would be for you to get that money*?”, with responses ranging from 1 (*I could easily raise the money*) to 4 (*I don’t think I could raise the money*). This item has been demonstrated to be a good index of financial hardship [18].

### 4.2. Parental Adjustment

The parent adjustment subscale of the Parent and Family Adjustment Scale—developmental disability version (PAFAS-DD) was used to measure parental adjustment difficulties. The PAFAS-DD has 30 items measuring parenting and family adjustment [19] on a scale from 0 (None at all) to 3 (Very much/most of the time) with higher scores indicating a higher level of dysfunction within families. The Parental Adjustment subscale has five items that assess parents’ emotional adjustment at the time of the survey. Examples of questions include: “*I feel satisfied with my life*” or “*I cope with the demands of being a parent*”. PAFAS-DD Parental Adjustment subscale has been shown to be a reliable measure to assess parental adjustment difficulties with internal consistency found in previous studies ranging from *α* = 0.81; [11] to *α* = 0.82 [19]. PAFAS-DD has also been shown to have satisfactory construct and convergent validity [19].

### 4.3. Coercive Parenting

The Coercive Parenting subscale of the PAFAS-DD was used to measure participants’ levels of coercive parenting. The Coercive Parenting subscale is comprised of five items that describe parenting behaviors such as: “*I shout or get angry at my child when they misbehave*” or “*I get annoyed with my child*”. Parents indicated how true the statement is to their parenting practice on a scale from 0 (None at all) to 3 (Very much/most of the time). A higher score indicates more use of coercive parenting. This scale has been demonstrated to be a valid and reliable measure of coercive parenting in families of children with DD. Composite internal consistency found for this subscale was 0.75, [19] and internal consistency was *α* = 0.73 [11].

### 4.4. Child Behavioral and Emotional Problems

Parents reported child behavioral and emotional problems using the Child Adjustment and Parenting Efficacy Scale—Developmental Disability (CAPES-DD) [20]. CAPES-DD has 30 items that describe different behavioral and emotional problems in children. Examples of items are: ‘*breaks or destroys things*’ (Behavioral problems) and ‘*seems fearful and scared*’ (Emotional problems). Parents indicate how accurately the problems describe their children by rating on a scale from 0 (‘*Not true of my child at all*’) to 3 (‘*True of my child very much, or most of the time*’). The CAPES-DD yields three scores. The Behavioral Problems score, Emotional Problems Score and Total Problems score. CAPES-DD has consistently been found to have good internal consistency with Cronbach’s alpha ranging between (*α* = [0.80–0.90]) for both the subscales and the total score [11,21]. CAPES-DD total problems scale also correlates strongly with the total behavior problems scale of the Developmental Behavior Checklist for both Primary Carer and Under-4 versions [21].

### 4.5. Analytic Strategies

Repeated measures ANOVA was adopted to calculate the change of scores across three time points. The relationship between parental adjustment and coercive parenting was explored using latent growth modelling (LGM) in which intercepts represent baseline score and slopes represent latent change over time. LGM allows researchers to explore the growth of individual constructs while simultaneously examining the relationship between several constructs. Bi-directional relationships were estimated with the error covariances, and the unidirectional relationships were estimated with path coefficients. A comparative fit index (*CFI*) value ≥ 0.95 and the root means square error approximation (*RMSEA)* value ≤ 0.08 indicates a good fit.

To estimate the contribution of change in parental adjustment and change in coercive parenting to subsequent child behavioral and emotional problems, hierarchical multiple regression was conducted. Demographic variables of families and child behavioral/emotional problems at baseline were controlled at Step 1 and Step 2 before changes in parental adjustment and coercive parenting (Time 1–Time 2) were entered at Step 3 to predict Time 3 child’s emotional and behavioral problems. The analyses were undertaken using AMOS and R software for statistic computing.

## 5. Results

### 5.1. Missing Data Analysis

The analysis of missing data indicated there was 18.7% missingness in total. Little’s MCAR test was not significant (*X**^2^ =* 175,818.70, *df* = 190,183, *p* > 0.05). The maximum likelihood estimation method was used to handle missing data.

### 5.2. Change in Parental Adjustment, Parenting Behaviors, and Child Behaviors

Table 2 shows the mean score of PAFAS- Adjustment, PAFAS—Coercive, CAPES-DD- Behaviors and CAPES-DD Emotion at Time 1, Time 2, and Time 3. The PAFAS- Adjustment score of 5.75 (*SD =* 3.00) pre-intervention, decreased significantly (*F*(1,890) = 80.37, *p* < 0.05) to 5.11 (*SD* = 2.61) post-intervention, and was maintained at the 12-month follow-up period ending at 5.02 (*SD =* 2.56). A significant reduction in the Coercive parenting score was also observed from Time 1 to Time 2. At Time 1, PAFAS- Coercive score was *M =* 9.66 (*SD* = 2.61) which reduced significantly to *M =* 8.73 (*SD =* 2.27) at Time 2 (*F*(1,890) = 230.31, *p* < 0.05). This effect was maintained at Time 3 at *M* = 8.85, *SD* = 2.09 (*F*(1,890) = 157.20, *p* < 0.05).

To examine whether the change scores are different across different levels of intervention, analysis was conducted controlling for the level of intervention. Results showed no interaction effect between time and level of intervention for any of the variables either short-term or long-term.

Child behavior problems estimated with the CAPES-DD Behavior started at Time 1 at *M* = 23.04 (*SD* = 6.65) then significantly reduced to *M* = 21.61 (*SD* = 6.15) at Time 2 (*F*(1,890) = 116.27, *p* < 0.05). The comparison between Time 3 and Time 1 was also significant (*F*(1,890) = 153.74, *p* < 0.05) indicating that the effect was maintained at Time 3.

Child emotional problems measured by CAPES-DD Emotion was 5.04 (*SD* = 1.80) at Time 1 and reduced significantly to *M* = 4.73 (*SD* = 1.57) (*F*(1,890) = 42.95, *p* < 0.05). This change was maintained at Time 3 (*F*(1,890) = 14.75, *p* < 0.05).

### 5.3. The Association between Change in Parental Adjustment and Change in Coercive Parenting

The LGM model to test the correlation of changes between parental adjustment and coercive parenting is presented in Figure 1 and Table 3. The model fits the data well (*X^2^* = 29.508, *df* = 7, *p* < 0.05; *CFI* = 0.993, *RSMEA* = 0.060). At Time 1, parental adjustment was significantly and positively associated with coercive parenting (*covariance coefficient* = 0.40, *p* < 0.05), indicating that those who experienced more adjustment difficulties at Time 1 were more likely to use coercive parenting strategies and vice versa. The examination of the change scores showed that changes in parental adjustment throughout intervention were significantly and positively associated with the change in coercive parenting (*covariance coefficient* = 0.16, *p* < 0.05). This finding suggests that a reduction in parental adjustment difficulties was associated with a reduction in coercive parenting.

### 5.4. The Association between Changes in Parental Adjustment and Coercive Parenting to Change in Child Behavioral and Emotional Problems

The association between variables are presented in Table 4.

Prior to conducting the regression analysis, all assumptions for multiple regression including linearity, multivariate normality, multicollinearity, and homoscedasticity were conducted. Data met the assumptions required for multiple regression.

After accounting for demographics and child behavioral/emotional problems at Time 1, the change in coercive parenting throughout Intervention (Time 1–Time 2) significantly predicts the behavioral problems (β=0.10, p<0.01) and child’s emotional problems (β=0.11, p<0.01) at Time 3. However, changes in parental adjustment did not significantly predict either child emotional or behavioral problems at Time 3 (Table 5).

## 6. Discussion

This study sought to extend the literature by examining how changes in parental adjustment and parenting skills at key timepoints throughout SSTP intervention affect each other and subsequently influence children’s outcomes. To our knowledge, this is the first mechanism of change analysis of the SSTP intervention. Results of this study suggested two main findings: First, the changes in parental adjustment over the course of intervention were associated with the changes in coercive parenting such that a decrease in parental emotional adjustment was correlated with a decrease in coercive strategies used. These change processes appear to co-occur such that when there is a reduction in parental emotional adjustment, there is also a reduction in coercive parenting. Second, the decrease in coercive parenting but not parental emotional adjustment achieved via intervention significantly contributed to children’s behavioral and emotional performance at follow-up.

When examining the relationship between baseline performance and the trajectory of changes, we found that parents who used more coercive parenting and reported more emotional difficulties at baseline demonstrated greater changes over the course of intervention. Such findings were consistent with and supported previous studies of a behavioral parenting intervention, suggesting that families with more problems at baseline respond better to intervention [22,23]. Families with more problems at baseline may have greater scope for growth thus demonstrating better progress than families with fewer initial concerns of which little improvement was needed.

Research conducted on families of children with DD in the past two decades has highlighted that an effective intervention for children with DD needs to address parental distress in order to bring about change in children. This argument is based on evidence to the link between parental adjustment difficulties and children’s outcomes [8,12]. In this study, we hypothesized that as parent’s emotional difficulties decrease, children’s behavioral and emotional problems will mutually deescalate. In contrast to our hypothesis, we found that change in parental adjustment throughout intervention was significantly associated with the change in coercive parenting, but its increase or decrease over the course of intervention did not contribute directly to the increase or decrease of subsequent child behavioral and emotional problems.

To explain this finding, we need to understand the SSTP program’s model of change. SSTPs and Triple P programs are built on the foundation of self-regulation principles. According to Sanders [24], self-regulation refers to the ability to change one’s own behavior and become an independent problem-solver by gaining the skills necessary to achieve one’s personal goals. In social cognitive theory, self-regulation is viewed as an essential process through which individuals can guide their behavior through changing circumstances over time. It involves modulating thought, affect, behavior, or attention with specific mechanisms and supportive meta-skills. As a result, SSTP training encourages parents to identify their goals, plan and self-select the most appropriate strategies to manage their emotions, and conduct self-evaluations addressing changes if needed [25]. The parents are thus able to attribute changes to their own behavior and effort rather than the child’s difficulties. Furthermore, self-evaluation might help parents become more aware of their own behavior and better able to assess situations before responding to them. By self-regulating, parents may have been able to distinguish their own emotions from their children’s difficulties and avoid passing on emotional disturbances to their children [26].

The findings of this study have implications for the development of parenting programs that support families of children with developmental disabilities; suggesting that parenting skills are key to influencing children’s behavior. Meanwhile, it is also important to highlight that although parental adjustment was not directly correlated with change in children’s behaviors, change in parental adjustment throughout the intervention, actually fostering healthy functioning parenting. As parents experience less emotional distress, they might be able to focus on building a positive, nurturing relationship with their children, resulting in a decrease in child behavioral and emotional problems

The finding of a non-significant relationship between change in parental adjustment and subsequent child outcomes nevertheless needs to be interpreted with care. It is possible too that some aspects of parental adjustment, such as the presence of parental mental health problems (anxiety and depression) might take longer to recover and change in response to reductions in problematic child behavior. There might also be a floor effect for problems, as most parents in our sample did not experience elevated adjustment difficulties hence a relatively small effect size was observed. Future studies with multiple data collection points and with clinically elevated samples might be useful to understand the cumulative and interactionally dependent between parental adjustment and child behavior change over time.

Finally, the mechanism for testing relationships between variables was restricted to three-time point data and was based on single parent reports which might require further validation. As Rutter [27] has suggested, to answer causal questions about development, there is a need for integrating longitudinal data in experimental intervention. More data points would allow a more accurate trajectory of change across individuals using LGM. Additional assessment time points would also allow a more definitive conclusion of the causal relationships which would enhance our understanding of the mechanisms by which changes in parent-related variables produce changes in child outcomes.

This study suggests that positive parenting skills are the most salient intervention ingredient driving the change in child behaviors, and the continued focus on building parenting capacities and parenting skills is justified. Although conducting a moderator analysis was not the primary focus of this study, our findings of baseline effect on family changes over the course of intervention are valuable in pointing to subgroups that might benefit the most from SSTP intervention. Our regression model findings also indicate several potential moderators (baseline status of families, DD with or without ASD, parental education, and family financial hardship) that can impact families’ capacity to change through intervention. Future studies with adequate sample sizes and more advanced analytic techniques could conduct moderator-mediator analysis to explore how mechanisms of change might vary by moderator groups.

## 7. Conclusions

Using a community-based sample of SSTP roll-out, this study emphasizes the significant role of parenting behaviors in improving children’s outcomes and suggests that developing warm, positive relationships between parents and their children should continue to be a priority in evidence-based parenting programmes for parents of children with DD. Additionally, this study also highlights the importance of promoting parental emotional well-being throughout intervention in order to mitigate the tendency to engage in coercive or negative parenting practices that are detrimental to children’s development.

## Figures and Tables

**Figure 1 ijerph-19-13200-f001:**
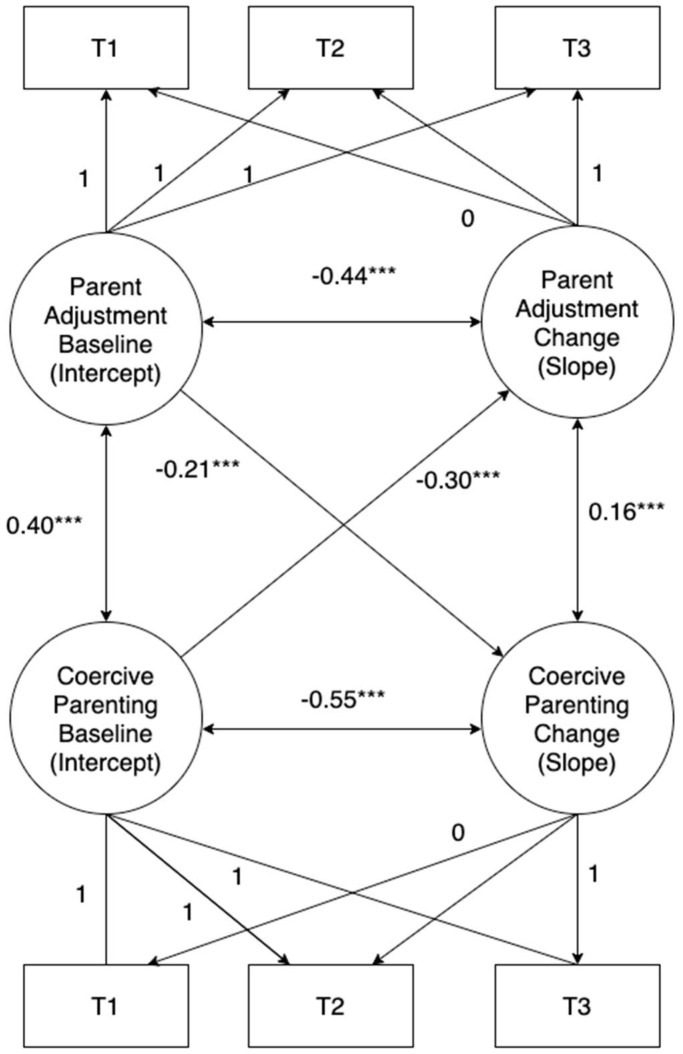
Latent growth Model of Intervention Outcomes. Note: *** *p* < 0.001.

**Table 1 ijerph-19-13200-t001:** Demographic characteristics.

Characteristic	Frequency	%
Child gender		
Male	687	77.10
Female	204	22.90
Child age (*Mean* and *SD*)	4.98 (2.65)	
Diagnosis with or without ASD		
With ASD	684	76.77
Without ASD	207	23.23
Caregivers’ relationship to the child		
Mother	781	87.65
Father	83	9.32
Grandparents or other relatives	27	3.03
Financial hardship		
Financial hardship	115	12.91
No financial hardship	717	80.47
Level of SSTP Intervention		
Level 2	248	27.83
Level 3	57	6.40
Level 4	381	42.76

Note: The accumulate percentage might not equal to 100% due to missing data.

**Table 2 ijerph-19-13200-t002:** *Mean*, *SD* of Variables.

	Time 1	Time 2	Time 3	Main Effect of Time	
	*M* *(SD)*	*M* *(SD)*	*M* *(SD)*	Time 1–Time 2	Time 1–Time 3
Parental adjustment	5.75 (3.00)	5.11 (2.61)	5.02 (2.56)	*F*(1,890) = 80.37, *p* < 0.05	*F*(1,890) = 97.72, *p* < 0.05
Coercive parenting	9.66 (2.61)	8.73 (2.27)	8.85 (2.09)	*F*(1,890) = 230.31, *p* < 0.05	*F*(1,890) = 157.20, *p* < 0.05
Child Behavior	23.04 (6.65)	21.61 (6.15)	21.22 (5.78)	*F*(1,890) = 116.274 *p* < 0.05	*F*(1,890) = 153.74, *p* < 0.05
Child Emotion	5.04 (1.80)	4.73 (1.57)	4.84 (1.50)	*F*(1,890) = 42.95, *p* < 0.05	*F*(1,890) *=* 14.75, *p* < 0.05

**Table 3 ijerph-19-13200-t003:** Estimate, Standard error of Coefficient and *p*-values of slopes and intercepts association.

	Estimate	Standard Error	*p*-Value
Parental Adjustment Intercept -> Coercive Parenting Slope	−0.210	0.22	<001
Coercive Parenting Intercept -> Parental Adjustment Slope	−0.299	0.31	<0.001
Parental Adjustment Intercept <-> Coercive Parenting Intercept	0.396	0.088	<0.05
Parental Adjustment Slope <-> Coercive Parenting Slope	0.161	0.276	<0.001

**Table 4 ijerph-19-13200-t004:** Cross-correlation between variables.

		(1)	(2)	(3)	(4)	(5)	(6)	(7)
1	T1 Child Behavior	1						
2	T3 Child Behavior	0.77 **						
3	T1 Child Emotion	0.559 **	0.431 **					
4	T3 Child Emotion	0.440 **	0.581 **	0.634 **				
5	T1 Coercive Parenting	0.328 **	0.182 **	0.172 **	0.094 **			
6	T2 Coercive Parenting	0.286 **	0.234 **	0.156 **	0.177 **	0.731 **		
7	T1 Parental Adjustment	0.321 **	0.298 **	0.261 **	0.257 **	0.346 **	0.306 **	
8	T2 Parental Adjustment	0.292 **	0.306 **	0.238**	0.305 **	0.278 **	0.390 **	0.760 **

Note: ** *p* < 0.01.

**Table 5 ijerph-19-13200-t005:** Hierarchical multiple regression to predict child Behavior and Emotion problems at Time 3.

	Time 3 Behavior Problems			Time 3 Emotion Problems		
	B	SE	β	B	SE	β
Step 3						
Child gender	0.06	0.34	0.00	0.07	0.11	0.02
Child age	−0.24	0.05	−0.11 ***	−0.04	0.02	−0.08 *
With or without ASD	0.85	0.35	0.06 **	0.42	0.11	0.12 ***
Level of intervention	0.10	0.15	0.02	−0.02	0.05	−0.01
Parent education	0.19	0.08	0.06 *	0.00	0.03	0.01
Marital status	0.16	0.32	0.01	0.11	0.10	0.03
Financial hardship	0.60	0.38	0.04	0.33	0.12	0.09 *
Time 1 Behavior/Emotion	0.65	0.02	0.75 ***	0.45	0.03	0.57 ***
T1–T2 Change in Parental Adjustment	−0.03	0.07	−0.01	0.00	0.02	0.00
T1–T2 Change in Coercive Parenting	0.32	0.08	0.10 ***	0.09	0.03	0.11***
R^2^	0.61 ***			0.38 ***		

* *p* < 0.05; ** *p* < 0.01; *** *p* < 0.001. ASD: Autism Spectrum Disorder.

## Data Availability

The data presented in this study are available on request from the corresponding author. The data are not publicly available due to ethics commitment of data security.

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
