# Peer review of "Predictors of Change in Stepping Stones Triple Interventions: The Relationship between Parental Adjustment, Parenting Behaviors and Child Outcomes"

_ijerph, 2022, doi:10.3390/ijerph192013200_

Round 1

Reviewer 1 Report

The current manuscript describes a study that intends to analyze predictors of change in an intervention for parents of children with disabilities. I believe that the study of predictors and moderators of change in behavioral interventions is a relevant topic because it is a necessary step to develop personalized prevention. However, the manuscript has several limitations, most notably, there are problems related to the consistency between the aims and the analytical approach. Also, there is some information in the method that is missing.

Here are some comments related to the manuscript limitations:

Abstract and introduction

The abstract is too short, it could be useful to provide more details about the study, particularly, including some of the results would be more informative.

I believe that the order of the information in the introduction can be rearranged. Is important to note that the manuscript title focuses in Stepping Stones Triple P (SSTP), so it is expected that the first paragraph describes it with detail then describe the results of previous studies.

I also believe that more detailed information about the differences between the classical version of the triple p program and the SSTP should be mentioned in the introduction.

Authors mention that “Several meta-analyses of randomized controlled trials”, however they only cite three references. I believe that three systematic reviews cannot be considered “several”

Methods

Which were the inclusion and exclusion criteria? Please describe how many families were invited to participate in the study, how many agree to participate, and how many were actually analyzed.

In participants it states that parents were reached using facebook, however, in procedures it seems that they were referred by a physician. To me is uncertain how the participants were recruited.

I can’t see why the authors use repeated measures ANOVA, latent growth modelling and multiple regression as analytic procedures. I believe that performing latent growth modelling would be enough to assess both hypotheses. From my point of view, the aims  were testing the associations at the slope level between parental adjustment and coercive parenting (objective 1) and the association of both slopes (parental adjustment and coercive parenting) with the slope of behavioral and emotional problem (objective 2), therefore only a latent growth model with three latent slopes and three latent intercepts is needed.

Additionally, in the terminology used in Figure 1 is confusing, instead of using “baseline” and “change”, use common use terms such as “intercept” and “slope”. This took me a lot of time to decipher what the authors did performed in their analysis.

In this case, as every measured score was treated as an observable outcome, is common that fit measures reach a good fit. Therefore, what matters the most in theses analysis is testing their main hypothesis such as the slope level associations.

I believe that the diagram of the latent growth model is neat and informative, however more information is required, such as, the standard error of the coefficients, and the results of the statistical analysis. Additionally, it would be useful to see R squared values.

It seems that authors used a community sample for this analysis, however it is unclear if there was any missing data. Please describe what was the approach to handle missing data. Also, please state which estimator was used. Did you assess multivariate normality?

Did you consider any confounding variables?

Did you consider potential site differences or differences in the method of intervention delivery?

Author Response

Response to Reviewer 1

Reviewer 1

The current manuscript describes a study that intends to analyze predictors of change in an intervention for parents of children with disabilities. I believe that the study of predictors and moderators of change in behavioral interventions is a relevant topic because it is a necessary step to develop personalized prevention. However, the manuscript has several limitations, most notably, there are problems related to the consistency between the aims and the analytical approach. Also, there is some information in the method that is missing.

Here are some comments related to the manuscript limitations:

Abstract and introduction

The abstract is too short, it could be useful to provide more details about the study, particularly, including some of the results would be more informative.

  • The abstract now contains more information about the outcomes

I believe that the order of the information in the introduction can be rearranged. Is important to note that the manuscript title focuses in Stepping Stones Triple P (SSTP), so it is expected that the first paragraph describes it with detail then describe the results of previous studies.

  • The Intro has since been revised and information was rearranged according to Reviewer’s suggestion. The Information of SSTP was move upfront

I also believe that more detailed information about the differences between the classical version of the triple p program and the SSTP should be mentioned in the introduction.

  • More detail regarding the differences between Standard Triple P and SSTP was added on page 1 of the manuscript. See text as below:

SSTP is designed to assist families with pre-adolescent children with disabilities who are experiencing or are likely to experience behavior problems. In addition to the core Triple P strategies, the program incorporates a number of additional disability-related components to reflect the additional challenges faced by parents of children with disabilities as well as a focus on community living and family support movements (such as Being part of the community). These include: (1) Identifying additional factors that are more likely to contribute to the development of behaviour problems in persons with disabilities (e.g., the accidental reward for stopping disliked activities). (2) Incorporating other behaviour change strategies from the disability literature (such as setting up an activity schedule). (3) Developing additional protocols to deal with self-injurious behaviour, repetitive behaviours, and pica that are more prevalent among children with disabilities. (4) Modifying wording and examples in parenting materials to make them more accessible and sensitive to parents of children with disabilities.”

Authors mention that “Several meta-analyses of randomized controlled trials”, however they only cite three references. I believe that three systematic reviews cannot be considered “several”

  • The word several is now removed from the paragraph.

Methods

Which were the inclusion and exclusion criteria? Please describe how many families were invited to participate in the study, how many agree to participate, and how many were actually analyzed.

In participants it states that parents were reached using facebook, however, in procedures it seems that they were referred by a physician. To me is uncertain how the participants were recruited.

  • Clarification is now added to the Recruitment section. Please see text as below:

Participants in this study were 891 parents and caregivers living in the Australian states of Victoria and Queensland and enrolled in the Mental Health of Young People with Developmental Disabilities (MHYPeDD) research study. These were caregivers of a child aged between 2 and 12 years who were recruited via a variety of pathways e.g., posters, brochures, and newsletters prepared by the project team and disseminated by their current service provider, a project-specific Facebook page, direct contact from the project team or via their child's school. Interested parents then provided evidence of a of diagnosis of DD provided by a suitable professional such as a psychiatrist, psychologist, speech pathologist, neurologist or pediatrician.”

I can’t see why the authors use repeated measures ANOVA, latent growth modelling and multiple regression as analytic procedures. I believe that performing latent growth modelling would be enough to assess both hypotheses. From my point of view, the aims were testing the associations at the slope level between parental adjustment and coercive parenting (objective 1) and the association of both slopes (parental adjustment and coercive parenting) with the slope of behavioral and emotional problem (objective 2), therefore only a latent growth model with three latent slopes and three latent intercepts is needed.

  • We would like to thank the Reviewer a chance to clarify the rationale behind our decision for choosing the analytic procedures described above. Objective 2 explores whether changing coercive parenting and changing parental adjustment over time will result in changes in children's outcomes. In doing so, we also wanted to control for other factors that may contribute to children's outcomes such as level of intervention, ASD diagnosis, or child gender, child age, parental education, marital status, and financial hardship, all of which have been found to influence intervention outcomes for children with developmental disabilities.

As the reviewer points out, both slopes of parental adjustment (parental adjustment and coercive parenting) can be associated with behavioural and emotional problem slopes using a latent growth model with three domains (parental adjustment, coercive parenting, and child outcomes). It is worth pointing out, however, that we only have three data points with the current data. In the case of multi-domain latent growth models with more than three variables and only three data points, model identification becomes challenging. As the degree of freedom decreased, the model required a larger number of extra constraints to be valid, especially when we controlled for other demographic variables. This makes multi-domain latent growth with three variables unsuitable for our purposes. Regression analysis, on the other hand, provides an efficient and reliable way to achieve our objective and is free from measure occasion requirements.   

Additionally, in the terminology used in Figure 1 is confusing, instead of using “baseline” and “change”, use common use terms such as “intercept” and “slope”. This took me a lot of time to decipher what the authors did performed in their analysis.

  • The words Baseline and Change were used as these terms are more familiar to the clinician audiences. We however have added technical terms in bracket.

In this case, as every measured score was treated as an observable outcome, is common that fit measures reach a good fit. Therefore, what matters the most in theses analysis is testing their main hypothesis such as the slope level associations.

  • The level of associations between parental adjustment and coercive parenting’s slopes are demonstrated in Figure 1. Though the association between the slopes of parental adjustment and coercive parenting with children outcomes’ slopes was not able to be achieved due to the limitation of the data set (as explained above), we believe that the regression analysis provides sufficient evidence for the association between change in parental adjustment and coercive parenting during the intervention and subsequent change in children’s emotional outcomes and their behaviors, thus meeting our objective 2

I believe that the diagram of the latent growth model is neat and informative, however more information is required, such as, the standard error of the coefficients, and the results of the statistical analysis. Additionally, it would be useful to see R squared values.

  • Thank you for this feedback, we have added more information re standard error of coefficients and slope level associations to the manuscript (Table 4).

It seems that authors used a community sample for this analysis, however it is unclear if there was any missing data. Please describe what was the approach to handle missing data. Also, please state which estimator was used. Did you assess multivariate normality?

  • Additional information regarding missing data and the procedure to handle missing these data was added to the Result section on Page 6 - text as below:

The analysis of missing data indicated there was 18.7% missingness in total. Little's MCAR test was not significant (X2 =175818.70, df=190183, p>.05). Maxi- mum likelihood estimation method was used to handle missing data.

  • Multivariate normality together with other assumption for multiple regression analysis such as multicolinearity, homoscedasticity were was assessed prior to conducting of regression analysis. The tests showed that data meet the assumption with no sign of muticolinearity, the errors between observed and predicted values were normally distributed (multivariate normality) and that the variance of error terms are similar across the values of the independent variables (Homoscedasticity). Due to the limit space for the manuscript, however we did not report the outcomes of the assumption testing. We have added a sentence into the Result section to acknowledge that that the assumptions were tested prior to the conducting of main analysis on page 9 or text as below:

“Prior to conducting the regression analysis, all assumptions for multiple regression including linearity, multivariate normality, multicollinearity, and homoscedasticity were conducted. Data meet the assumptions for multiple regression.

Did you consider any confounding variables? Did you consider potential site differences or differences in the method of intervention delivery?

  • Other cofounding variables were not accounted for in the analysis for Objective 1 as we mainly interested in the association of change between parental adjustment and coercive parenting. However for Objective 2, we did account for other confounding variables when exploring the factors that determine children emotion and behaviour outcomes as shown in Table 4 including the differences in level of intervention intensity, ASD diagnosis, children age, gender, parental education, marital status and family financial harship. These are the variables that have been found in the literature of Stepping Stones Triple P to impact program outcomes. Site differences was not accounted for as large body of evidence has shown that Triple P the effect of Triple P was adequate across different cultures and social contexts, within Australia and oversea.

Reviewer 2 Report

Q1:The introduction and discussion sections need to be supplemented with references from the last three years, especially those from 2020 and 2022.

Q2: in order to improve the readability of this article, I recommend the author add subheadings in the introduction section to summarize the main contents of paragraphs.

Q3:In Table 3 and Table 4, I recommend the author to explain the abbreviation concepts in the figure and make notes at the bottom of the figure, such as CAPES- DD,PAFAS , and ASD.

Q4:  Attention should be paid to the format of the article. For example, where should the reference No. 35 under the fourth paragraph of the discussion part be placed?

Q5: In the discussion part, the author needs to use appropriate theory to explain the findings of the research.

Author Response

Response to Reviewer 2

Q1:The introduction and discussion sections need to be supplemented with references from the last three years, especially those from 2020 and 2022.

  • More recently published articles were now cited in the Introduction and discussion.

Q2: in order to improve the readability of this article, I recommend the author add subheadings in the introduction section to summarize the main contents of paragraphs.

  • Subheadings were added according to Reviewer’s suggestions

Q3:In Table 3 and Table 4, I recommend the author to explain the abbreviation concepts in the figure and make notes at the bottom of the figure, such as “CAPES- DD,PAFAS , and ASD”.

  • We went ahead and changed all the abbreviations in Table 3 and 4 into interpretable terms so they are consistent with other tables. We also added the explanation to foodnote of Table 4 to explain for the term ASD.

Q4:  Attention should be paid to the format of the article. For example, where should the reference No. 35 under the fourth paragraph of the discussion part be placed?

  • We have carefully revised the manuscript and place all references accordingly.

Q5: In the discussion part, the author needs to use appropriate theory to explain the findings of the research.

  • The self-regulation framework and social cognitive theory were highlighted in the discussion to explained for our findings on page 12.

“To explain this finding, we need to understand the SSTP program's model of change. SSTPs and Triple P programs are built on the foundation of self-regulation. According to Sanders [31], self-regulation refers to the ability to change one's own behavior and become an independent problem-solver by gaining the skills necessary to achieve one's personal goals. In social cognitive theory, self-regulation is viewed as an essential process through which individuals can guide their behavior through changing circumstances over time. It involves modulating thought, affect, behavior, or attention with specific mechanisms and supportive meta-skills. As a result, SSTP training encourages parents to identify their goals, plan and self-select the most appropriate strategies to manage their emotions, and conduct self-evaluations addressing changes if needed [32]. The parents are thus able to attribute changes to their own behavior and effort rather than the child's difficulties. Also, self-evaluation might help parents become more aware of their own behavior and better able to assess situations before responding to them. By self-regulating, parents may have been able to distinguish their own emotions from their children's difficulties and avoid passing on emotional disturbances to their children [33] .”

Round 2

Reviewer 1 Report

I have three main concerns that regarding to this manuscript, and I believe are unchanged in this current version. 

1. The clarity of the manuscript is limited, and sometimes is hard to understand.

2. It seems that for some of the analysis, data was unsuitable, therefore, authors needed to perform several types of analysis, most of them are non necesarily compatible in their assumptions and interpretation. 

3. Is important to understand that sit effects in real settings and provider heterogeneity is a reality in psychosocial interventions, and, in an observation study like this one, they must be accounted to ensure the robustness of findings. 

Author Response

Response to Reviewer 1

I have three main concerns that regarding to this manuscript, and I believe are unchanged in this current version. 

  1. The clarity of the manuscript is limited, and sometimes is hard to understand. We thank the Reviewer for this feedback. The Introduction was revised in accordance with the Reviewer’s specific suggestions that included clearly laying out the theoretical background, rationales as well as the aims and hypotheses of the study.
  2. It seems that for some of the analysis, data was unsuitable, therefore, authors needed to perform several types of analysis, most of them are non necesarily compatible in their assumptions and interpretation. 

We would like to thank the Reviewer for the feedback. We however believe that our data is suitable for the research question and the analysis chosen in this study is appropriate to answer the research question posed. 

Our study has 2 hypotheses: (1) the decrease in parental adjustment difficulties over the course of intervention will be associated with the simultaneous decrease in coercive parenting behaviors and (2) the decrease in parental adjustment difficulties and coercive parenting from pre- to post-intervention will predict subsequent decrease in child behavioral and emotional problems at follow-up.

First, in term of data, our data contains information of parental adjustment difficulties, coercive parenting and also children’s outcomes at pre intervention, post intervention and 6-month follow-up which is sufficient to answer the research question and test the hypotheses posed. Our data, therefore, is suitable for the study purpose.  

Second, for the analysis chosen, we used Latent growth modelling to test the simultaneously change between parental adjustment and coercive parenting during the intervention (Hypothesis 1) and we used regression to test their effect on subsequent child outcomes (Hypothesis 2), which are both reliable methods to test each of the Hypothesis. 

Latent growth modelling has become widely used to examine the simultaneously trajectory of change between different domains in recent years while the regression analysis has been used for many years as a reliable method to analyze the predictor of change. Separately and combined, these methods have been used in numerous public health studies in the past and our study is not the first one that combine these types of analysis to test for the process of change in intervention. 

See studies that adopted Regression Analysis to examine predictor of change: Feig, E.H., Harnedy, L.E., Celano, C.M. et al. Increase in Daily Steps During the Early Phase of a Physical Activity Intervention for Type 2 Diabetes as a Predictor of Intervention Outcome. Int.J. Behav. Med. 28, 834–839 (2021). https://doi.org/10.1007/s12529-021-09966-0

OR 

Allen, T. A., Lam, R. W., Milev, R., Rizvi, S. J., Frey, B. N., MacQueen, G. M., ... & Quilty, L. C. (2019). Early change in reward and punishment sensitivity as a predictor of response to antidepressant treatment for major depressive disorder: a CAN-BIND-1 report. Psychological medicine49(10), 1629-1638.

AND

See Moreland, A.D., Felton, J.W., Hanson, R.F. et al. The Relation Between Parenting Stress, Locus of Control and Child Outcomes: Predictors of Change in a Parenting Intervention. J Child Fam Stud 25, 2046–2054 (2016). https://doi.org/10.1007/s10826-016-0370-4 who adopted similar approach using both Latent change analysis and Regression to answer similar research question,  testing for the relation between Parenting Stress and Locus of Control during intervention and subsequent influence on child outcomes

  1. Is important to understand that site effects in real settings and provider heterogeneity is a reality in psychosocial interventions, and, in an observation study like this one, they must be accounted to ensure the robustness of findings. 

While site differences in intervention delivery are possible, we chose not to explore these site difference as moderators for the following reasons. (1) Efforts were made to minimise site differences by ensuring that all practitioners received identical competency and accreditation-based training, (2) the interventions used the same practitioner and parent resources and also (3) because prior research on training outcomes in regular service delivery settings has shown few differences in training outcomes for practitioners from different disciplines, countries, or level of programs being used.

Reviewer 2 Report

1. In the section of Relations between parental adjustment, children outcomes and parenting behaviors in families of children with DD, the research of Baker, McIntyre, Blacher, Crnic, Edelbrock and Low does not support the transactional relationship of parental stress and child behavioral and emotional problems.

2. Please illustrate that how the introduction supports hypothesis 1 and 2.

3. Table 4 is not clear. Please write the variable name completely and present Table 4 clearly.

Author Response

Response to Reviewer 2

  1. In the section of Relations between parental adjustment, children outcomes and parenting behaviors in families of children with DD, the research of Baker, McIntyre, Blacher, Crnic, Edelbrock and Low does not support the transactional relationship of parental stress and child behavioral and emotional problems.

 We would like to thank the Reviewer for the detail, the section has been rewritten and the reference is no longer presented.

  1. Please illustrate that how the introduction supports hypothesis 1 and 2.

The Introduction was revised so that it clearly lay out the theoretical background and rationales that link to the aims and hypotheses of the study.

  1. Table 4 is not clear. Please write the variable name completely and present Table 4 clearly.

The variables in Table 4 are now presented clearly.